# Screening conditions and constructs for attempted genetic transformation of *C. elegans* by *Agrobacterium*

**Eleanor C. Warren**[1,2], **Andre E. X. Brown**[1,2*], **Karen S. Sarkisyan**[1,2*]

**1** MRC Laboratory of Medical Sciences, London, United Kingdom, **2** Institute of Clinical Sciences, Faculty of Medicine, Imperial College London, London, United Kingdom

* andre.brown@lms.mrc.ac.uk (AEXB); k.sarkisyan@lms.mrc.ac.uk (KSS)

## Abstract

Manipulating gene expression within a model organism is important for reverse genetic experimentation, and while techniques to generate transgenic *C. elegans* are available, they are optimised for creating individual lines. The ability to create libraries of genetically modified animals using *C. elegans* as a model would make new types of experiments possible and would speed up studies of animal physiology. Here, we describe a range of constructs designed to establish a high-throughput method of *C. elegans* transformation mediated by gene transfer from *Agrobacterium.* We demonstrate that *C. elegans* are able to survive on *Agrobacterium* as a sole food source, and screen conditions for *Agrobacterium-mediated* transformation in this organism. While we do not achieve routine gene transfer from *Agrobacterium* to *C. elegans,* we suggest that this technique has potential following further optimization. The success of the approach would enable rapid and high-throughput transformation of *C. elegans,* providing an improvement on currently available methods. Here we provide details of optimization conditions tested, and a useful resource of T-binary constructs for use by the scientific community.

## Introduction

The nematode *Caenorhabditis elegans* is an extensively used model organism for genetic analysis, with major discoveries stemming from work on this organism. Despite the popularity of *C. elegans* as a research model, methods of transgenesis of this organism are limited. The most common method of transgenesis in *C. elegans* is microinjection of DNA into the gonad, where DNA rearranges to form inherited concatemers, or can be triggered to integrate into the genome [1]. While this technique is widely adopted and effective, it traditionally allows for the injection of only a single animal at a time, which is laborious and thus incompatible with the generation of large libraries of genetically modified animals.

**Data availability statement:** All relevant data are within the paper and its Supporting Information files.

**Funding:** All work is funded by the MRC laboratory of Medical sciences (https://www.ukri.org/councils/mrc/). The Synthetic Biology Group (KS & ECW) is funded by the MRC Laboratory of Medical Sciences (UKRI MC-A658-5QEA0). The Behavioural Genomics group (AEXB) is funded by the MRC Laboratory of Medical Sciences (UKRI MC-A658-5TY30). The funders did not play a role in the study design, data collection or analysis, decision to publish or preparation of the manuscript.

**Competing interests:** The authors have declared that no competing interests exist.

Biolistic bombardment has also been employed to integrate transgenes within the *C. elegans* genome. This technique involves accelerating DNA-coated microparticles toward a population of worms [2,3]. Although antibiotic resistance markers can be delivered to aid identification of successful transformants, the efficiency of transgenesis using this method remains low [4]. Electroporation, a technique which is widely used for the introduction of exogenous DNA in other organisms, has been successfully applied to *C. elegans* for the delivery of dsRNA for RNAi silencing [5], however, attempts to achieve transgenesis of *C. elegans* through electroporation have resulted in only transient, non-inheritable gene expression [5,6], primarily due to challenges in targeting DNA to the germline.

Recently, a higher-throughput library transgenesis approach, termed TARDIS, has been reported [7]. This technique circumvents the low throughput nature of microinjection by transforming a single animal with a library of genetic elements, creating heritable arrays, which are subsequently integrated into the genome to yield genetically diverse individuals. TARDIS relies on the ability of *C. elegans* to join recombinant DNA into large and unstable extrachromosomal arrays, a process of which is not well understood mechanistically, and cannot be fully controlled by the researcher.

While genome-wide resources of constructs for use *in C. elegans*, such as a library of constructs for the expression of GFP-tagged proteins, have been created [8], these resources remain under-utilised due to the laborious nature of transforming each construct individually. A simpler and more direct high-throughput transgenesis protocol would be beneficial for research of animal physiology, enabling rapid creation of large libraries of genetically engineered *C. elegans* for high-throughput screening.

*Agrobacterium*-mediated transformation is the most widely used method for generating transgenic plants, and has shown success in delivering genes into organisms from other kingdoms, including yeasts [9], filamentous fungi [10], algae [11], human cells [12], and sea urchin embryos [13]. This demonstrates the potential for *Agrobacterium*-mediated transformation as a tool for genetic engineering across kingdoms, including transgenesis of whole animals. While the value of applying *Agrobacterium*-mediated transformation to *C. elegans* has been previously recognized [14], there are no records of this technique successfully producing transformants in this organism.

In nature, *Agrobacterium* introduces genes into plant genomes, which leads to tumour formation and the biosynthesis of amino acids and opines, creating a beneficial environment for the bacteria to thrive [15]. Because the DNA delivery happens in a sequence-agnostic manner, this ability to integrate genes into host genomes has been harnessed for genetic engineering by replacing tumour-inducing genes with genes of interest: any DNA sequence placed between the two 25-base-pair repeats, T-DNA borders, can be integrated into the target genome.

*Agrobacterium*-mediated transformation requires the expression of virulence genes which encode the machinery essential for the transfer and integration of T-DNA into the genome of the target organism. For the purposes of genetic engineering, a binary system has been developed where the T-DNA is present in a T-binary

vector, and the virulence genes are expressed from a second plasmid called the Vir helper plasmid (Fig 1a). Induction of virulence genes by plant wound-induced phenolic compounds, such as acetosyringone, results in transfer of a single-stranded copy of plasmid-encoded T-DNA via a type IV secretion system into the target cell [16]. This system has been optimised for a variety of different plants and other species [12,17–20].

The discovery that *Agrobacterium* is able to achieve T-DNA transfer and genome integration in human cells [12] and sea urchin embryos [13] suggests that the processes of DNA delivery and integration are conserved between plants and animals, indicating that this approach could be applied more wildly to other animal models. For *Agrobacterium*-mediated transformation to produce heritable germline transformants in *C. elegans,* the T-DNA would require access to the germ-line. This could be achieved through direct contact of *Agrobacterium* with germ cells, oocytes or sperm, or through DNA shuttling from tissues accessed by *Agrobacterium*, such as the intestine or vulva (Fig 1b). While heritable germline trans-formation would be ideal for the creation of transgenic animals, *Agrobacterium*-mediated transformation may also result in transient somatic transformation of the cells directly exposed to *Agrobacterium*, the cuticle, the lumen of the pharynx, the intestine, the vulva and the anus (Fig 1b). Here, we attempted to utilise *Agrobacterium* to introduce transgenic DNA into *C.*

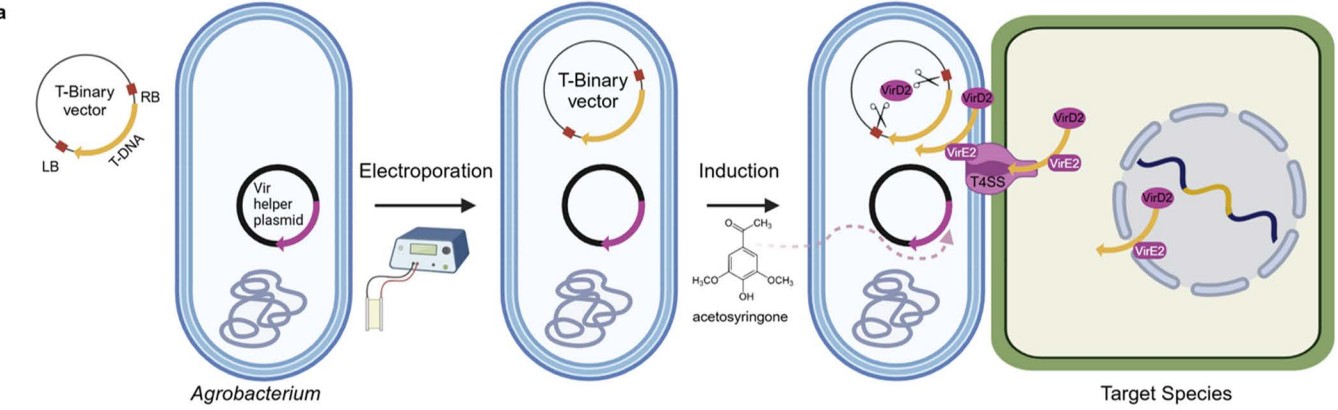

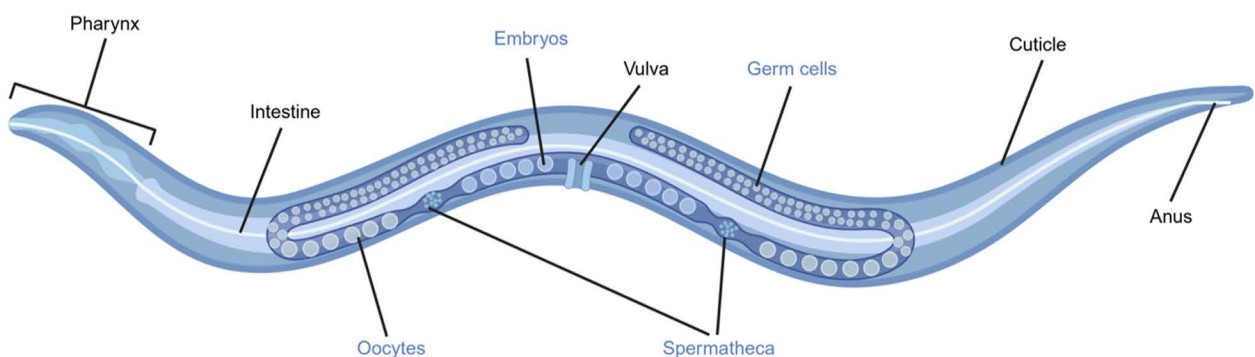

**Fig 1. *Agrobacterium*-mediated transformation of *C. elegans*.** (a) T-binary system. DNA to be integrated into the target genome is cloned between T-DNA borders (LB, and RB) in a T-binary vector, which is introduced to *Agrobacterium* by electroporation. Induction of virulence genes on a Vir helper plasmid is triggered with acetosyringone, resulting in cleavage of T-DNA and its delivery to the nucleus of the target species for insertion in the genome. (b) Anatomy of a *C. elegans* hermaphrodite, indicating reproductive components (labelled in blue), and suggested sites for *Agrobacterium* access (labelled in black).

*elegans* either transiently or in a heritable manner in order to create a transgenesis approach useful for high-throughput screening.

## Results

### C. *elegans* survive with *Agrobacterium* as a food source

For *Agrobacterium*-mediated transgenesis to be possible, infection of animals with live *Agrobacterium* must take place. We initially examined the ability of *C. elegans* to feed on *Agrobacterium* as a sole food source and assessed both length and speed of worms fed either on *E. coli* (OP50)*,* the diet routinely fed to *C. elegans* in the laboratory, or on *Agrobacterium tumefaciens* strain LBA4404 (referred to as *Agrobacterium*). Adult *C. elegans* fed on *Agrobacterium* were shorter in length (Fig 2a, p < 0.0001, Unpaired Mann-Whitney test), while speed was not significantly different (Fig 2b, Unpaired Mann- Whitney test). *C. elegans* were able to survive on *Agrobacterium,* with survival comparable to *C. elegans* on *E. coli* (S5 Fig)*.* *C. elegans* were able to produce viable progeny on *Agrobacterium,* with eggs visible 3 days after refeeding

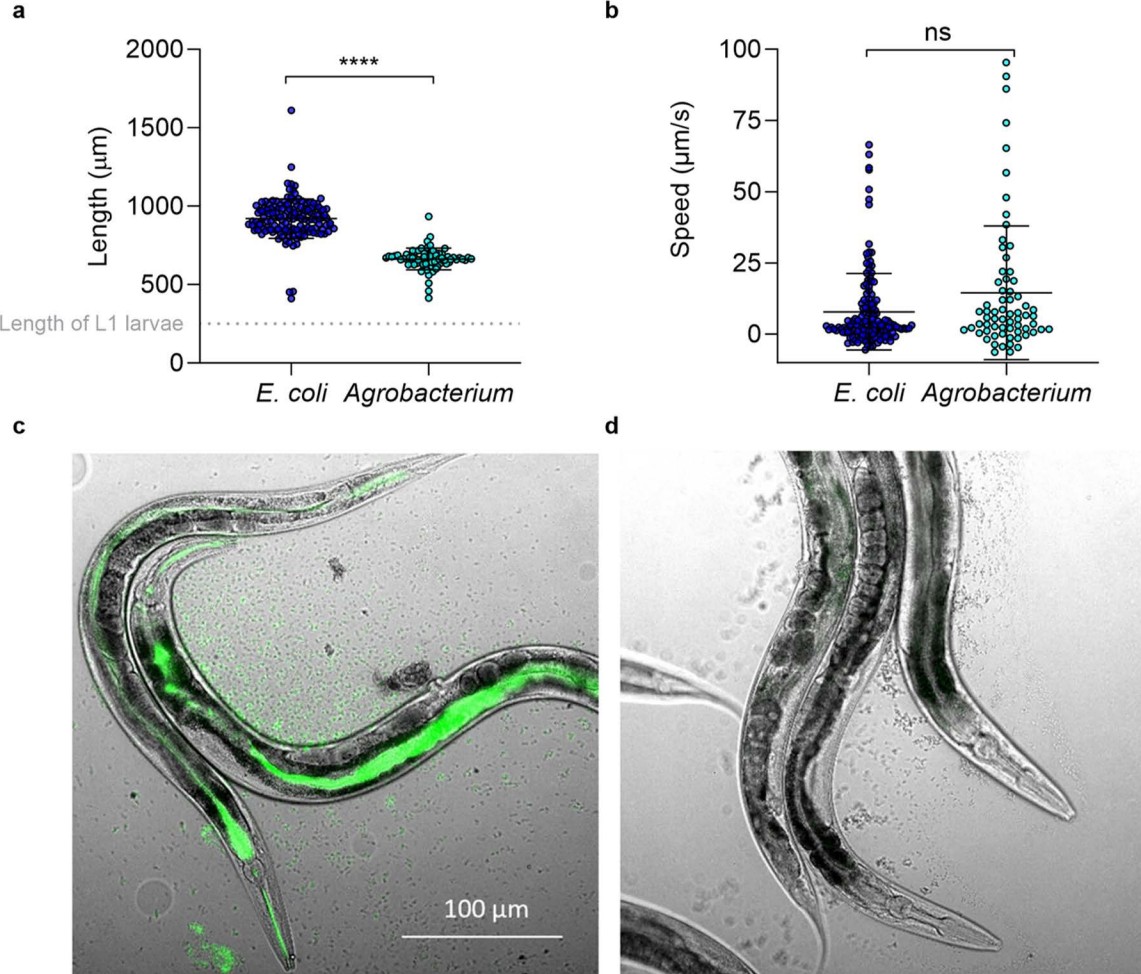

**Fig 2. Appearance of *C. elegans* fed on *Agrobacterium*.** Adult *C. elegans* fed on *Agrobacterium* or *E. coli* were monitored for (a) length and (b) speed (Mann–Whitney test, n > 72. Error bars represent mean +/- SD). Widefield microscopy of *C. elegans* fed (c) GFP expressing *Agrobacterium* or (d) non fluorescent *Agrobacterium*.

L1s, and F1 progeny hatching 4 days after refeeding (S5 Fig). Eggs from the F1 and F2 generation were laid earlier in *C. elegans* fed *Agrobacterium* (induced or uninduced) compared to *C. elegans* fed *E. coli* (p ≤ 0.01 Kruskal–Wallis test with Dunn's post hoc test), however, no attraction to *Agrobacterium* over *E. coli* was observed from a chemotaxis assay (S5 Fig, p > 0.05, Unpaired Mann-Whitney test). Additionally, we fed *C. elegans Agrobacterium* expressing GFP and observed fluorescence within the lumen of the pharynx and within the intestine of the worm both in the P0 generation and in the progeny (Fig 2c).

## Attempt to generate fluorescent *C. elegans* through the introduction of fluorescent constructs using *Agrobacterium*-mediated transgenesis

With the aim of introducing detectable transgenic gene expression into *C. elegans* using *Agrobacterium*-mediated transformation, we fed worms *Agrobacterium* containing T-binary vectors with T-DNA comprising transcriptional units designed to drive the expression of fluorescent proteins in *C. elegans*. Transcriptional units designed with a *C. elegans* specific promoter (*Prpl-28*) and 3'UTR (*let-858* 3' UTR) expressing either wrmScarlet or mTagBFP2 (referred to herein as BFP) were introduced into T-binary vectors and delivered to *Agrobacterium* via electroporation. If these constructs are incorporated into worms, this promoter should drive expression ubiquitously in the progeny, alternatively insertion into somatic cells would be expected to produce mosaic animals with fluorescence in specific targeted tissues [21].

Successfully transformed *Agrobacterium* were induced to express virulence genes, with the aim of triggering transport, insertion, and expression of T-DNA from within *C. elegans* cells. *C. elegans* from different life stages (from L1 to adult) were exposed to induced *Agrobacterium*, and fluorescence was monitored both in the parental generation (P0) and in the progeny (F1). Strong levels of autofluorescence were observed in worms fed *Agrobacterium*, however no additional fluorescence was reproducibly observed in worms fed *Agrobacterium* designed to introduce wrmScarlet (Fig 3a) or BFP (Fig 3b) expression.

On two separate occasions, worms fed *Agrobacterium* with T-DNA designed to introduce wrmScarlet expression exhibited remarkable fluorescence (Fig 3c). This pattern of atypically bright fluorescence was also observed on 10 occasions with a construct (*Pmyo3*::mCherry::*unc-54*), where the fluorescent gene was designed to contain introns to preclude any expression of the fluorescent gene by the *Agrobacterium* (Fig 3c). The conditions used to obtain these results are highlighted in S1 Table. On these twelve occasions, this phenotype was displayed by all *C. elegans* observed, including in the P0 and the F1 generation (S3 Fig), however this bright fluorescence was not consistently reproducible across different experiments even when these conditions were reproduced. This bright fluorescence appeared to localise to the pharynx, the vulva, and two regions approximately 100 μm anterior and posterior to the vulva. Additionally, fluorescence was sometimes observed in the intestine. In order to assess the background fluorescence of *Agrobacterium,* cells with and without T-binary vectors were monitored for fluorescence (S2 Fig). Presence of T-DNA designed for the expression of fluorescent proteins in *C. elegans* did not lead to increased fluorescence of *Agrobacterium*, both in constructs with introns (encoding mCherry) or without introns (encoding wrmScarlet), indicating that fluorescent proteins with *C. elegans*-specific promoters were not expressed by *Agrobacterium. Agrobacterium* did, however, display higher autofluorescence when grown on certain media types, such as Todd Hewitt (TH) as compared to LB-Nematode Growth Media agar (LB-NGM) (S2 Fig).

## Phenotyping *C. elegans* behaviour fed *Agrobacterium* containing test constructs

As an additional metric to assess *Agrobacterium*-mediated gene delivery to *C. elegans,* we selected a range of genes that – if exogenously expressed in *C. elegans* – would be expected to result in severe paralysis. Genes chosen for inclusion within T-DNA were the neuropeptide *flp-1* (dna1959), and Amyloid-β (Aβ1–42) (dna1977), both of which have been demonstrated to cause inhibited movement or paralysis when expressed in *C. elegans* [22,23], as well as MmTX1, a peptide from coral snake venom which has been reported to paralyse worms when applied topically [24]. These genes were included as transcriptional units within T-DNA in *Agrobacterium*, with the sequence for MmTX1 either alone (dna1948) or with signal

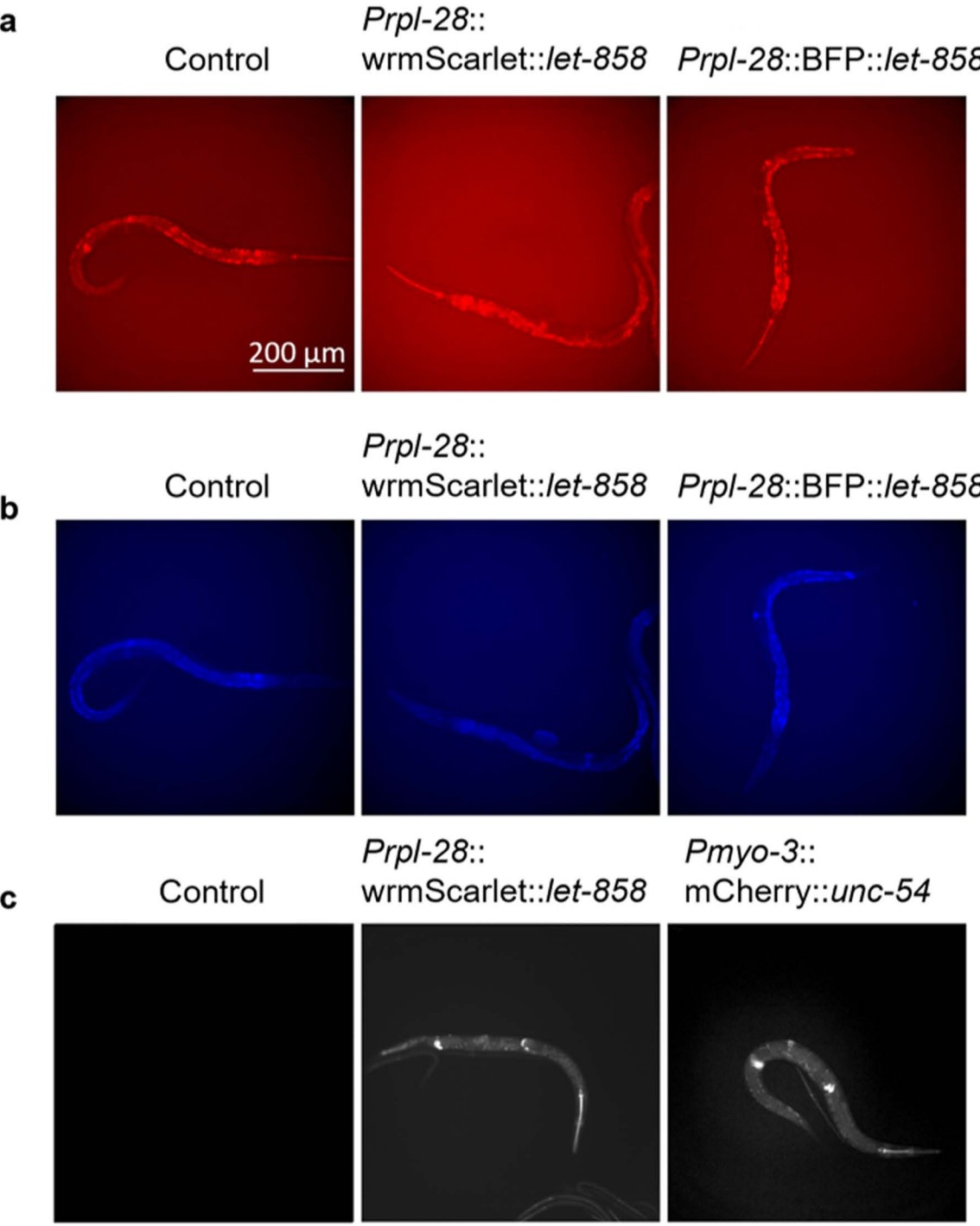

**Fig 3. Autofluorescence of *Agrobacterium* infected *C. elegans*.** *C. elegans* fed on *Agrobacterium* control or *Agrobacterium* containing T-binary vectors (*Prpl-28*::wrmScarlet::*let-858* or *Prpl-28*:BFP::*let-858*) induced to transfer T-DNA to host, were monitored for fluorescence using widefield microscopy in the (a) DsRed channel or (b) blue channel. (c) Under certain conditions *C. elegans* fed *Agrobacterium* with the T-binary vector *Prpl-28*::wrmScarlet::*let-858* or *Pmyo3*::mcherry::*unc-54* showed strong fluorescence in specific regions.

peptide sequences, *clec-1* (dna1947), or *Spp1*(dna1946) [25]. Controls of wrmScarlet (WS) (dna1401), BFP (dna1621) or no plasmid were included. *C. elegans* larvae were fed on these strains of induced *Agrobacterium*, and at adulthood their behaviour was monitored to assess the frequency of forward motion and frequency of pausing as a read out of paralysis. We found no significant differences in forward motion (Fig 4a), or pausing (Fig 4b) under any condition tested, suggesting that the genes of interest were not successfully expressed, at least not at levels sufficient to paralyse worms.

## Screening conditions for *Agrobacterium* induction and coincubation with *C. elegans*

We screened several parameters of the transformation protocol trying to achieve successful gene delivery, including strain of *Agrobacterium* and T-binary vector used, induction conditions, and conditions for *C. elegans* coincubation with *Agrobacterium* (Fig 5).

Since infection by *Agrobacterium* is likely necessary for bacterial attachment and T-DNA transfer, we also investigated whether some of the available *C. elegans* mutant strains would be potentially susceptible to infection. Strains tested

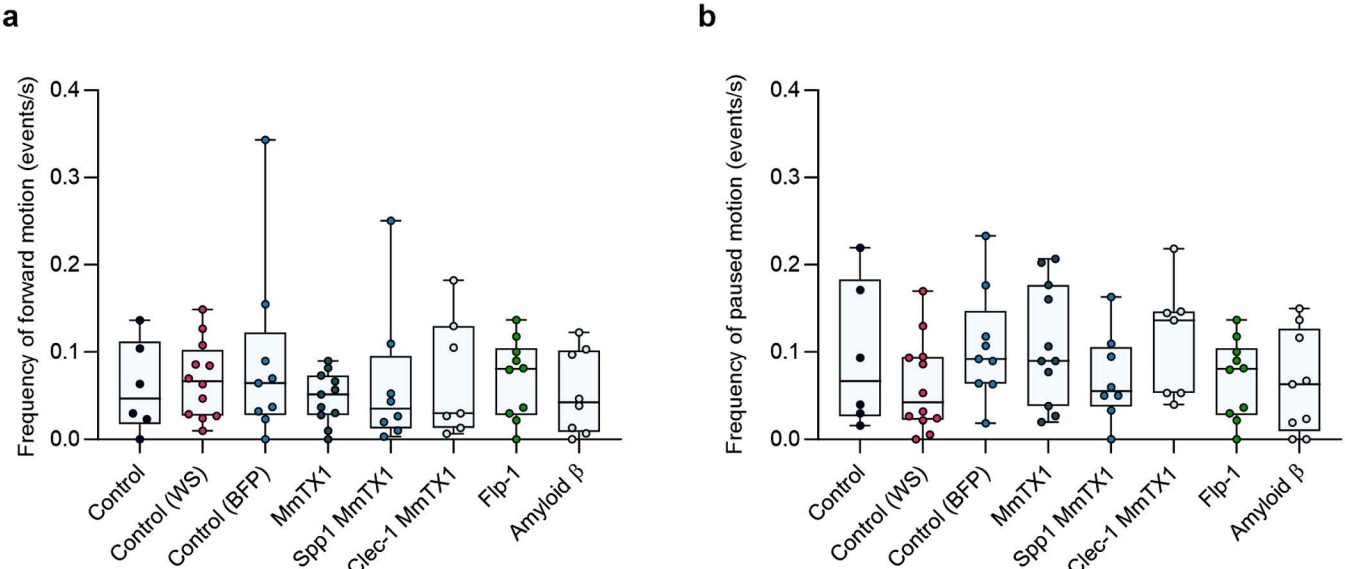

**Fig 4. Locomotion of *C. elegans* fed on *Agrobacterium* control strain, or strains containing T-binary vectors engineered to paralyse *C. elegans*.** (a) Frequency of forward motion, (b) frequency of paused motion (Kruskal–Wallis test with Dunn's post hoc test, n ≥ 6, boxes represent IQR, line shows median value). Each independent replicate represents an average of 3 worms per well.

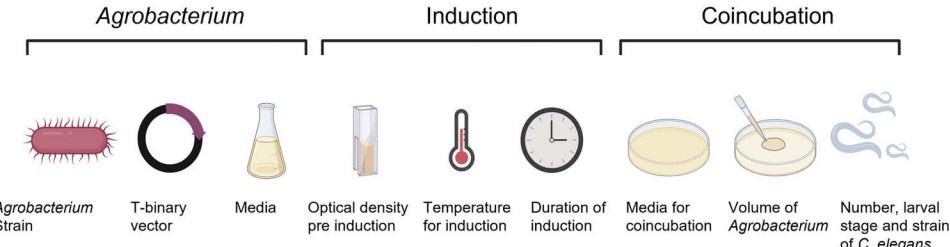

**Fig 5. Schematic of the transformation protocol parameters varied during testing for *C. elegans* transformation using *Agrobacterium*.** Expanded details of experimental conditions are provided in S1 Table.

included a mutant defective in grinding bacteria (*phm-2(ad597) I.*), which has previously been shown to allow live bacteria to accumulate in the intestine [26], a mutant with a disrupted cuticle integrity (*bus-17(e2800) X.*) [27], and a mutant with intestinal invaginations (*bbln-1(mib70) X.*) [28]. Given that the vulva may serve as an entry point for *Agrobacterium*, we also explored *Agrobacterium*-mediated transgenesis in *C. elegans* using the *him-5(e1490) V.* strain, which exhibits a high incidence of males. Our aim was to facilitate mating between male worms and hermaphrodites, thereby enhancing agro-bacterial entry through this route and potentially increasing access to the germline. However, none of the tested conditions or their combinations resulted in successfully transformed *C. elegans* in our hands. A summary of all the conditions tested is shown in S1 Table.

### T-binary vectors for use in *C. elegans*

A range of T-binary vectors for use in *C. elegans* have been generated (Table 1). DNA sequences are available (S1 Fig). We employed modular cloning vectors designed for Golden Gate construction of multigene constructs [29,30]. We selected Modular cloning vectors MoClo level 1 position 1 (pICH47732), position 2 (pICH47742) and position 3 (pICH47751) (for singular transcription units) or MoClo level 2 (pAGM4723) (for multiple transcription units) which contain origins of replication for *E. coli* and *Agrobacterium,* left and right borders for T-strand transfer, and Type IIS restriction sites compatible with Golden Gate cloning [29,30]. All T-DNA sequences were designed with well documented promoter and 3'UTR pairs for ubiquitous expression, *Prpl-28* and *let-858* 3'UTR [31], *Peft-3* and *tbb-2* 3'UTR [32], and *Prps-27* and *unc-54* 3'UTR [33] or targeted expression *Pmyo-3* and *unc-54* 3'UTR [34]. A range of fluorescent proteins were chosen: wrmScarlet, BFP, mCherry, GFP and eYFP, to enable visual identification of transformed organisms. Transcription units containing the genes for *flp-1*, Amyloid-β (Aβ1–42) and MmTX1 (with and without secretion signals) were constructed, cloned into the backbones alone or in combination with wrmScarlet. Finally T-DNA sequences containing a transcription unit designed to express a puromycin resistance gene [35] were created, and are available in combination with other transcription units in MoClo level 2.

## Discussion

*Agrobacterium* have previously been shown to transfer genes to a range of organisms under a variety of conditions, suggesting that *Agrobacterium*-mediated transformation could be successfully applied to a wide range of species. In this study we have designed a range of plasmids for use in *Agrobacterium*-mediated transformation of *C. elegans,* a technique which could vastly increase the speed and throughput of transgenesis of nematodes. We have systematically varied conditions for *Agrobacterium* growth, induction, and coincubation with *C. elegans,* however, have not successfully generated routine transgenesis of nematodes using this method. Here we are reporting our results and test conditions, and are sharing the plasmids created for *Agrobacterium*-mediated transformation.

  *C. elegans* is a widely used model, with over 1000 labs worldwide studying this organism. Transgenic *C. elegans* are most commonly created via individual microinjection of worms [36]. Currently, large-scale forward genetic screens are possible where mutagenesis or RNAi is used to disrupt gene function, and phenotypes are subsequently identified. However, the ability to generate vast numbers of *C. elegans*-expressing libraries of functional transgenes*,* through a technique such as *Agrobacterium*-mediated transformation, would enable large scale screens of mutants expressing endogenous, or exogenous proteins or peptides [37,38]. Given the high efficiency of *Agrobacterium* transformation in plant systems, as demonstrated by the ability to achieve near-simultaneous delivery of over 20 unique T-DNAs to a given plant cell [39] we believe *Agrobacterium* holds significant potential for use in *C. elegans* transformation.

  Here we have failed to generate transgenic *C. elegans* using *Agrobacterium*-mediated transgenesis, however in some conditions we have observed intense fluorescence in specific regions of the worms, such as the vulva and two regions anterior and posterior to the vulva (Fig 3c and S4 Fig). While this fluorescence is noticeably brighter, it is located in comparable regions where autofluorescence has been observed in control *Agrobacterium* lacking a T-binary vector (Fig 3a).

**Table 1. Constructs for testing *Agrobacterium*-mediated transformation.**

| | Summary of T-binary vectors | | | | | | | | | |
| | Transcription unit (position 1) | | | Transcription unit (position 2) | | | Transcription unit (position 3) | | | Backbone |
| | Gene | Promoter | 3'UTR | Gene | Promoter | 3'UTR | Gene | Promoter | 3'UTR | |
|---|---|---|---|---|---|---|---|---|---|---|
| dna1270 | mCherry | *Pmyo-3* | *unc-54* | | | | | | | L1 |
| dna1401 | wrmScarlet | *Prpl-28* | *let-858* | | | | | | | L1 |
| dna1627 | wrmScarlet | *Peft-3* | *tbb2* | | | | | | | L1 |
| dna1631 | wrmScarlet | *Prps-27* | *unc-54* | | | | | | | L1 |
| dna1640 | wrmScarlet | *Prpl-28* | *let-858* | | | | | | | L2 |
| dna1621 | BFP | *Prpl-28* | *let-858* | | | | | | | L1 |
| dna1624 | BFP | *Peft-3* | *tbb2* | | | | | | | L1 |
| dna1628 | BFP | *Prps-27* | *unc-54* | | | | | | | L1 |
| dna1638 | BFP | *Prpl-28* | *let-858* | | | | | | | L2 |
| dna1622 | eYFP | *Prpl-28* | *let-858* | | | | | | | L1 |
| dna1625 | eYFP | *Peft-3* | *tbb2* | | | | | | | L1 |
| dna1629 | eYFP | *Prps-27* | *unc-54* | | | | | | | L1 |
| dna1641 | eYFP | *Prpl-28* | *let-858* | | | | | | | L2 |
| dna1623 | GFP | *Prpl-28* | *let-858* | | | | | | | L1 |
| dna1626 | GFP | *Peft-3* | *tbb2* | | | | | | | L1 |
| dna1630 | GFP | *Prps-27* | *unc-54* | | | | | | | L1 |
| dna1642 | GFP | *Prpl-28* | *let-858* | | | | | | | L2 |
| dna1941 | | | | wrmScarlet | *Prpl-28* | *let-858* | | | | L1 P2 |
| dna1942 | | | | | | | PuroR | *Prpl-28* | *let-858* | L1 P3 |
| dna1986 | | | | wrmScarlet | *Prpl-28* | *let-858* | PuroR | *Prpl-28* | *let-858* | L2 |
| dna1987 | | | | | | | PuroR | *Prpl-28* | *let-858* | L2 |
| dna1984 | flp-1 | *Prpl-28* | *let-858* | | | | | | | L1 |
| dna1959 | flp-1 | *Prpl-28* | *let-858* | wrmScarlet | *Prpl-28* | *let-858* | | | | L2 |
| dna1960 | flp-1 | *Prpl-28* | *let-858* | wrmScarlet | *Prpl-28* | *let-858* | PuroR | *Prpl-28* | *let-858* | L2 |
| dna1975 | Amyloid Beta | *Prpl-28* | *let-858* | | | | | | | L1 |
| dna1976 | Amyloid Beta | *Prpl-28* | *let-858* | | | | | | | L2 |
| dna1977 | Amyloid Beta | *Prpl-28* | *let-858* | wrmScarlet | *Prpl-28* | *let-858* | | | | L2 |
| dna1978 | Amyloid Beta | *Prpl-28* | *let-858* | wrmScarlet | *Prpl-28* | *let-858* | PuroR | *Prpl-28* | *let-858* | L2 |
| dna1639 | MmTX1 | *Prpl-28* | *let-858* | | | | | | | L1 |
| dna1951 | MmTX1 | *Prpl-28* | *let-858* | | | | | | | L2 |
| dna1948 | MmTX1 | *Prpl-28* | *let-858* | wrmScarlet | *Prpl-28* | *let-858* | | | | L2 |
| dna1945 | MmTX1 | *Prpl-28* | *let-858* | wrmScarlet | *Prpl-28* | *let-858* | PuroR | *Prpl-28* | *let-858* | L2 |
| dna1939 | spp1-MmTX1 | *Prpl-28* | *let-858* | | | | | | | L1 |
| dna1949 | spp1-MmTX1 | *Prpl-28* | *let-858* | | | | | | | L2 |
| dna1946 | spp1-MmTX1 | *Prpl-28* | *let-858* | wrmScarlet | *Prpl-28* | *let-858* | | | | L2 |
| dna1943 | spp1-MmTX1 | *Prpl-28* | *let-858* | wrmScarlet | *Prpl-28* | *let-858* | PuroR | *Prpl-28* | *let-858* | L2 |
| dna1940 | clec1-MmTX1 | *Prpl-28* | *let-858* | | | | | | | L1 |
| dna1950 | clec-1-MmTX1 | *Prpl-28* | *let-858* | | | | | | | L2 |
| dna1947 | clec-1-MmTX1 | *Prpl-28* | *let-858* | wrmScarlet | *Prpl-28* | *let-858* | | | | L2 |
| dna1944 | clec-1-MmTX1 | *Prpl-28* | *let-858* | wrmScarlet | *Prpl-28* | *let-858* | PuroR | *Prpl-28* | *let-858* | L2 |

We have not observed this same pattern of autofluorescence in *C. elegans* fed *E. coli* (S4 Fig). Using GFP-expressing *Agrobacterium* we demonstrated that the *Agrobacterium* localise within the pharynx and intestine of *C. elegans* with no noticeable accumulation within these regions where we have observed fluorescence (Fig 2c). This indicates that the observed fluorescence is not caused by *Agrobacterium* accumulation, but may result from an interaction with the bacteria, complicating the interpretation of the red fluorescence as a potential T-DNA expression signal.

Genetic modification of *C. elegans* via traditional methods is well established, thus there are many well described regulatory elements for engineering targeted gene expression in this organism. For the generation of functional transcription units to be delivered as T-DNA from T-binary vectors we selected a range of promoters for targeted (*Pmyo-3*) [39], or ubiquitous (*Peft-3*, *Prpl*-28 and *Prps*-27) expression [1,32,33]. Well established 3' UTRs were chosen for ubiquitous expression (*unc-54*, *tbb-2*) [1], or permissiveness for germline expression (*tbb-2*, *let-858*) [40,41]. Genes chosen for easy to observe phenotypes (either fluorescence or behavioural paralysis) were chosen for expression from T-DNA, and in order to aid identification of successfully transformed nematodes, a puromycin resistance gene was designed to be ubiquitously expressed from an additional modular transcription unit [35]. The constructs designed here (Table 1) are suitable for *in vivo* genetic transformation of *C. elegans* animals, or transformation of *ex vivo* cultured *C. elegans* germ cells and should provide a useful resource for optimizing *Agrobacterium*-mediated transformation of *C. elegans*.

Most *Agrobacterium*-mediated transformation protocols share common features such as induction of virulence genes with acetosyringone and co-cultivation within a narrow temperature range of 22–26°C [17,42–44]. However, *Agrobacterium*-mediated transformation with sea urchin embryos has been successful in the absence of phenolic inducer or defined induction media [13] and co-incubation of HeLa cells with *Agrobacterium* has been successfully carried out at 37°C [12], much higher than has been established for plant protocols. This suggests that optimal conditions for *Agrobacterium*-mediated transformation differ according to species, and indicates that modification of existing protocols could result in successful T-DNA transfer from *Agrobacterium* to *C. elegans*.

Following the discovery of *Agrobacterium-mediated* transformation, and widespread use of this technique for generating genetically modified plants, optimised *Agrobacterium* strains have been developed to enhance transformation efficiency. Since certain strains appear to have differing efficacies in different species [45], we selected a range of *Agrobacterium* strains for testing with *C. elegans*: LBA4404 [46], C58C1[47], GV3101 [48], AGLO [49], EHA105 and A281 [50]. This includes strains used in transformation of HeLa (C58C1) and sea urchin embryos (GV3101), however, transformation into *C. elegans* was unsuccessful with all of these strains. For future optimization of *Agrobacterium*-mediated transformation, modifications to widely used strains could be made by improving the attachment of *Agrobacterium* to host cells. Work into enhancing attachment of *Agrobacterium* to non-natural hosts has been carried out by engineering adhesins to the surface of *Agrobacterium* [51], applying similar techniques for attachment to *C. elegans* cells could result in successful infection and T-DNA transfer. Furthermore, strains of *Agrobacterium* with reporters for virulence gene induction such as *lacZ* [52] or GFP [53], could be beneficial for optimizing induction conditions.

In addition to the strains of potentially-susceptible *C. elegans* mutants tested here, other *C. elegans* strains could be worth investigating, such as immunocompromised mutants (e.g., *sek-1(ag1)* or *nsy-1(ag3)*) with reduced innate immunity [54]. Strains of *C. elegans* such as vitellogenin mutant (*vit-1(ok2616)*X) with reduced eggshell integrity [55] could be useful to enable *Agrobacterium* to access and infect the germline. Alternatively, exogenous chemicals such as caffeine [56] could be used to physically permeabilise the eggshell, allowing *Agrobacterium* to transform cells of the germ line, thereby enabling heritable genetic modifications.

Several other factors in the procedure for *Agrobacterium*-mediated transformation, aside from *Agrobacterium* and host strain, have been shown to influence gene transfer efficiency. Such essential conditions include, the pH of the co-cultivation medium [57,58], the cell density of *Agrobacterium* [58,59], and media used for both induction and co-cultivation [60]. We systematically altered these conditions along with other variables in the *Agrobacterium*-mediated transformation protocol (S1 Table), with the aim of developing a procedure compatible with *C. elegans*. While we didn't

succeed in developing an *Agrobacterium*-mediated transformation protocol for *C. elegans,* we hope sharing details of our optimization strategy may prove useful for others.

## Materials and methods

### Construct construction

Sequences encoding genes, promoters and terminators were ordered as gene fragments from Twist Biosciences, or Integrated DNA Technologies (IDT), and cloned by golden gate assembly into MoClo level one plasmids (pICH47732: position 1, pICH47742: position 2 or pICH47751: position 3) before assembly of multiple transcription units into MoClo level 2 (pAGM4723). Except for *Pmyo-3*::mCherry::*unc-54* in L1 which was generated using Gibson assembly with DNA amplified from pCFJ104 [34], and MoClo L1 (pICH47732).

### Bacteria strains and vectors

*Agrobacterium tumefaciens* strain LBA4404 (ThermoFisher Scientific, 18313015) was used as a host for all experiments shown in this paper. During screening, a selection of *Agrobacterium* strains were tested (EHA105, GV3101, A281, AGLO and C58), all strains were maintained with Streptomycin 100 µg/ml and Rifampicin 25 µg/ml. *Agrobacterium* competent cells were transformed with T-binary plasmids according to manufacturer instructions (ThermoFisher, Scientific 18313015), and selected for using 50 µg/ml Carbenicillin (Level 1 backbones) or 50 µg/ml Kanamycin (level 2 backbones). PCR using T-DNA specific primers was used to confirm successfully transformed *Agrobacterium* colonies. Untransformed *Agrobacterium* lacking any T-binary vector was used as a negative control.

### C. *elegans* strains

*C. elegans* strains N2, *phm-2(ad597) I.* (DA597), *bus-17(e2800) X.* (CB6081) and *him-5(e1490) V.* (CB4088) were obtained from the Caenorhabditis Genetics Center (CGC). *C. elegans* strain *bbln-1(mib70) X.* was generously provided by Mike Boxem (Utrecht University).

### *Agrobacterium*-mediated transformation

A standardized protocol is available at dx.doi.org/10.17504/protocols.io.5qpvo9emzv4o/v2. *Agrobacterium* was streaked to LB or Yeast Extract Peptone (YEP) agar plates containing appropriate antibiotics to maintain plasmid vectors (100 µg/ml streptomycin, 50 µg/ml Carbenicillin or Kanamycin and 25 µg/ml Rifampicin), and plates were incubated for 3 days at 28°C. A single colony of *Agrobacterium* was used to inoculate 20 ml YEP media (with appropriate antibiotics) which was incubated overnight at 28°C and 250-rpm. Overnight cultures were diluted 1:100 in 20 ml YEP media or AB sucrose media ((0.5% sucrose, 3% $K_2HPO_4$, 1% $NaH_2PO_4$, 1% $NH_4Cl$, 0.145% $MgSO_4$, 0.15 KCl, 0.01% $CaCl_2$, and 0.0025% $FeSO_4$-$7H_2O$, 2 mM $NaPO_4$)(with appropriate antibiotics), and these cultures were incubated at 28°C and 250-rpm until the *Agrobacterium* reached an OD600 of 0.3. *Agrobacterium* cells were pelleted by centrifugation (5 min 4000g), before resuspension in 1 ml induction medium containing acetosyringone (100 µM or 200 µM), and shaking at 50-rpm at room temperature for varying durations (S3 Fig). 0.5 ml of *Agrobacterium* in induction media was added directly to LB-NGM (or alternative) plates with appropriate antibiotics. A mixed population of worms (containing *C. elegans* at different life stages (from L1 to adult), were washed three times in M9 buffer before 15 worms were transferred, by pipetting, to plates with *Agrobacterium*. Alternatively, *C. elegans* were bleach synchronised (dx.doi.org/10.17504/protocols.io.2bzgap6) and either L1s were added to *Agrobacterium* directly, or worms were refed until L4 stage before washing three times in M9 buffer and adding to *Agrobacterium*. Plates were incubated at 20 degrees for 4–10 days, such that the progeny of the exposed *C. elegans* can be observed. *C. elegans* fluorescence was observed for 100s of nematodes directly on plates using an Olympus U-RFL-T microscope, or for 10's of worms on microscope slides using a Olympus IX70 fluorescence

microscope. To anaesthetise *C. elegans* to prevent them moving during microscopy, *C. elegans* were transferred to microscope slides with 3% agar pads and 1ul of 25 mM sodium azide.

Multiple different conditions were tested for *Agrobacterium*-mediated transformation (S1 Table). Multiple *Agrobacterium* strains (LBA4404, EHA105, GV3101, A281, AGL0, C58), and T-binary vectors were utilised. Various conditions for *Agrobacterium* induction were screened, such as the optical density (OD600) of *Agrobacterium* for induction, the density of *Agrobacterium* in induction medium, the concentration of the phenolic inducer acetosyringone, the duration of induction, and the presence of antibiotics. Multiple media types were tested for co-incubation of *Agrobacterium* and *C. elegans.* These were LB-NGM (1% tryptone, 0.5% yeast extract, 1% NaCl, 1mM $CaCl_2$, 1mM $MgSO_4$, 1mM $KPO_4$, 1.7% BioAgar, 5 µg/mL Cholesterol), LB (1% tryptone, 0.5% yeast extract, 1% NaCl, 1.5% agar), Low salt LB (1% tryptone, 0.5% yeast extract, 0.05% NaCl, 1.5% BioAgar), SOB (2% tryptone, 0.5% Yeast extract, 0.05% NaCl, 0.0186% KCl, 0.203% MgCl, 0.24% $MgSO_4$, 1.5% BioAgar), YEP (1% Bacto peptone, 1% Yeast extract 0.5% NaCl, 1.5% BioAgar), Terrific (2% Tryptone, 2.4% Yeast extract, 0.4% Glycerol, $KPO_4$, 1.7 mM $KH_2PO_4$, 7.2 mM $K_2HPO_4$, 1.5% BioAgar), 2x Yeast extract Tryptone (2xYT) (1.6% Tryptone, 1% Yeast extract, 0.5% NaCl, 1.5% BioAgar), Yeast Mannitol (YM) (0.04% Yeast extract, 1% mannitol 1.7 mM NaCl, 0.8mM $MgSO_4$, 2.2 mM $K_2HPO_4$, 1.5% BioAgar), liquid Induction Media (IM)(1% $NH_4Cl$, 0.145% $MgSO_4$, 0.15 KCl, 0.01% $CaCl_2$, and 0.0025% $FeSO_4$-$7H_2O$, 2 mM $NaPO_4$ (pH 5.6), 50 mM 2-(4-morpholino)-ethane sulfonic acid (MES), 0.5% glucose, 100 µM acetosyringone), Induction Media agar (IM + 1.5% BioAgar), Dulbecco's Modified Eagle Medium (DMEM Gibco +1.5% BioAgar), Yeast Peptone (YP) (2% Bacto peptone, 1% Yeast extract, 1.5% BioAgar). All media was supplemented with 5 µg/ml cholesterol. The concentrations of streptomycin and carbenicillin in these plates were varied, as was the presence of surfactant and the hydration of the plates. For LB-NGM plates, the brand of LB and Agar was varied as were the potassium phosphate salts added. Additionally, the strain of *C. elegans*, number of worms, and temperature of incubation were systematically varied.

### Behavioural feature extraction

A detailed protocol is available at dx.doi.org/10.17504/protocols.io.n92ldro9og5b/v1. Briefly, tierpsy tracker was used to obtain behavioural features for worms in wells of a 96 well plate [61–63]. We extracted features describing morphology, "length_50th" (median worm length), and movement, "speed_50th" (median worm speed), "motion_mode_forward_frequency" (frequency of forward motion in events/second) "motion_mode_pasued_frequency" (frequency of paused motion in events/second).

### *C. elegans* progeny, lifespan and chemotaxis assays

A detailed protocol is available at dx.doi.org/10.17504/protocols.io.e6nvwbx59vmk/v2. For progeny assays, approximately 50 synchronized L1 N2 *C. elegans* were transferred onto 90 mm LB-NGM plates seeded with 1 mL of either *E. coli* (OP50), uninduced *Agrobacterium*, or induced *Agrobacterium*, all normalized to an OD600 of 1.8. Plates were monitored every 24 hours to record the number of eggs and progeny visible per plate. Data collection continued until F2 generation eggs were observed (day 6).

For lifespan assays, approximately 50 synchronized L4 N2 *C. elegans* were transferred onto LB-NGM plates supplemented with 400 µM 5-Fluorodeoxyuridine and seeded with 1 mL of *E. coli* (OP50), uninduced *Agrobacterium,* or induced *Agrobacterium*, all normalized to an OD600 of 1.8. *C. elegans* were monitored daily, and individuals were classified as dead if they failed to respond to a gentle touch to the head using an eyebrow pick.

Chemotaxis assays were performed on 90 mm LB-NGM plates seeded with 100 µL of *E. coli* (OP50) or induced *Agrobacterium* at opposite ends of the plates. Both cultures were normalized to an OD600 of 1.8. Approximately 40 synchronized L4 *C. elegans* were placed at the center of each plate. After 4 hours, the number of worms on each bacterial lawn was recorded and the Chemotaxis Index (CI) was calculated as (number of worms on *E. coli* - number of worms on *Agrobacterium*/ total number of worms).

## Statistics

The distribution of all experimental data was tested using the D'Agostino and Pearson omnibus normality test. Two-tailed non-parametric Mann-Whitney tests were used to assess significance between independent samples from two groups not showing a Gaussian distribution. The Kruskal-Wallis test with Dunn's post hoc test was used to test significance between three or more independent groups of data not showing a Gaussian distribution. Statistical analysis was carried out using GraphPad Prism Software.**** indicates $p \leq 0.0001$, p ** indicates $p \leq 0.01$, * indicates $p \leq 0.05$. Where no significance is indicated in a figure, $p > 0.05$.

## Supporting information

**S1 Fig. Genbank sequence files for all constructs for testing *Agrobacterium*-mediated transformation.** DNA Sequence files for constructs described in Table 1.
(ZIP)

**S2 Fig. Autofluorescence of *Agrobacterium* and *C. elegans* on LB-NGM or Todd Hewitt media.** (a) Autofluorescence of *Agrobacterium* containing different T-binary vectors grown on LB-NGM or Todd Hewitt (TH) agar was assessed by pelleting a known number of cells and visualizing the pellet in an Eppendorf tube using widefield fluorescence microscopy. Autofluorescence of *C. elegans* fed *Agrobacterium* grown on LB-NGM or TH agar was visualized using widefield fluorescence microscopy, using the DsRed channel. (b) Quantification of autofluorescence from *Agrobacterium* and *C. elegans* grown on LB-NGM or TH agar (Mann–Whitney test, n = 4, error bars represent mean +/- SEM).
(TIF)

**S1 Table. Experimental parameters for attempted *C. elegans* transformation using *Agrobacterium*.** C. elegans fed *Agrobacterium* carrying T-binary vectors engineered for fluorescence expression in worms occasionally exhibited fluorescence in specific body regions, which appeared noticeably brighter than the typically observed autofluorescence. Experimental conditions associated with these phenotypes are highlighted in green, and representative images are displayed in S3 Fig.
(TIF)

**S3 Fig. Representative fluorescence microscopy images of *C. elegans* from the highlighted conditions in S1 Table.** Images were captured using widefield fluorescence microscopy in the DsRed channel. Wild-type worms without *Agrobacterium* treatment served as controls, with three representative images shown.
(TIF)

**S4 Fig. Fluorescence observed in *C. elegans* following exposure to *Agrobacterium*.** *C. elegans* fed *Agrobacterium* carrying T-DNA engineered for fluorescence expression occasionally exhibited atypically bright fluorescence in the vulva and in two regions approximately 100 μm anterior and posterior to the vulva. (a) Fluorescence was detected in the vulva (labeled with *), as well as in two lateral regions (labeled with x), following exposure to *Agrobacterium* strain LBA4404 containing either the *Pmyo-3*::mCherry::*unc-54* or *Prpl-28*::wrmscarlet::*let-858* T-binary vector. Control worms exposed to *Agrobacterium* lacking the T-binary vector are also shown. (b) Autofluorescence observed in *C. elegans* fed *E. coli* (OP50) or *Agrobacterium* lacking the T-binary vector is compared to the fluorescence observed in *C. elegans* fed *Agrobacterium* carrying the *Prpl-28*::wrmscarlet::*let-858* construct. Images were captured using widefield fluorescence microscopy in the DsRed channel and brightfield microscopy.
(TIF)

**S5 Fig. Comparison of *C. elegans* lifespan and fecundity on *Agrobacterium* and *E. coli.*** (a) Survival curves of *C. elegans* fed *E. coli* (OP50), uninduced *Agrobacterium*, or induced *Agrobacterium* (Kruskal–Wallis test with Dunn's post

hoc test, n = 6). (b) Quantification of eggs and progeny laid over time by *C. elegans* fed *E. coli* (OP50), induced *Agrobacterium* or uninduced *Agrobacterium* (Kruskal–Wallis test with Dunn's post hoc test, n = 6). (c) Chemotaxis index of *C. elegans* comparing *E. coli* (OP50) against induced *Agrobacterium* (Mann–Whitney t test, n = 6. Box represents IQR, line shows median value).
(TIF)

## Author contributions

**Conceptualization:** Andre E. X. Brown, Karen S. Sarkisyan.

**Data curation:** Eleanor C. Warren.

**Formal analysis:** Eleanor C. Warren.

**Funding acquisition:** Andre E. X. Brown, Karen S. Sarkisyan.

**Investigation:** Eleanor C. Warren.

**Methodology:** Eleanor C. Warren.

**Supervision:** Andre E. X. Brown, Karen S. Sarkisyan.

**Writing – original draft:** Eleanor C. Warren.

**Writing – review & editing:** Eleanor C. Warren, Andre E. X. Brown, Karen S. Sarkisyan.

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
