## [Decision Letter · Decision Letter 0]

9 Jan 2025

PONE-D-24-38923Screening conditions and constructs for attempted genetic transformation of C. elegans by AgrobacteriumPLOS ONE

Dear Dr. Warren,

Thank you for submitting your manuscript to PLOS ONE. After careful consideration, we feel that it has merit but does not fully meet PLOS ONE’s publication criteria as it currently stands. Therefore, we invite you to submit a revised version of the manuscript that addresses the points raised during the review process.

We look forward to receiving your revised manuscript.

Kind regards,

Ramachandran Srinivasan, Ph.D.

Academic Editor

PLOS ONE

Journal Requirements:

2. We are unable to open your Supporting Information file [Genbank.zip ]. Please kindly revise as necessary and re-upload.

Reviewers' comments:

Reviewer's Responses to Questions

**Comments to the Author**

1. Is the manuscript technically sound, and do the data support the conclusions?

Reviewer #1: No

Reviewer #2: Yes

Reviewer #3: Partly

Reviewer #4: Yes

2. Has the statistical analysis been performed appropriately and rigorously? 

Reviewer #1: No

Reviewer #2: I Don't Know

Reviewer #3: Yes

Reviewer #4: Yes

3. Have the authors made all data underlying the findings in their manuscript fully available?

Reviewer #1: No

Reviewer #2: Yes

Reviewer #3: Yes

Reviewer #4: Yes

4. Is the manuscript presented in an intelligible fashion and written in standard English?

Reviewer #1: Yes

Reviewer #2: Yes

Reviewer #3: Yes

Reviewer #4: Yes

5. Review Comments to the Author

Reviewer #1: This article explores the development of a novel technique for C.elegans transgene. Currently, the most common approach involves injecting DNA, protein, or RNA into the gonads of worm via microinjection to establish a transgenic line for genetic analysis. However, microinjection presents a high technical barrier and low efficiency. The authors propose an alternative approach using Agrobacterium to generate transgene, which is an intriguing idea. However, while the authors have conducted various attempts, the results indicate that transgene has not been successfully achieved. It remains unclear whether the method is inherently unfeasible or if further optimization of the experimental conditions could yield success. Key challenges include:

1. Uncertainty regarding whether the virulence protein in Agrobacterium is effectively expressed.

2. The host-specific nature of the Type IV secretion system (T4SS), raising doubts about whether Agrobacterium can deliver DNA and protein into worm cells via this pathway.

3. Ambiguity around whether the virulence protein can insert T-DNA into the worm genome.

After carefully reviewing the methods section, I suggest the following improvements:

1. Induce the expression of key virulence proteins in Agrobacterium by using the worm liquid culture method (refer to WormBook). Resuspend Agrobacterium in S-medium and include appropriate inducers to promote virulence gene expression.

2. Detect the expression of virulence genes using qPCR after inducing.

3. Use synchronized worms for transgene. Transfer L4-stage or young adult (YA) worms to induced-Agrobacterium culture from step 1, followed by overnight liquid culture. Subsequently, transfer the worms to NGM plates contain OP50.

4. Examine the localization of Agrobacterium in the parental P0 generation.

5. Screen for transgenic worms in the F1 progeny. Use fluorescent labeling to detect the presence of the transgene in subsequent generations.

Reviewer #2: Here the authors present largely negative data, as well as resources and suggestions for further research, which could contribute to the application of Agrobacterium-mediated transformation in C. elegans. Publication of negative results is an important contribution to the scientific process, preventing others from wasting time and resources by repeating the same experiments, as well as guiding future research.

I have no major concerns, but do have some minor comments to help improve the manuscript:

1. There are still quite a few typos and style errors that need to be fixed prior to publication.

2. Statistics are missing for the data presented in Fig. 2.

3. Displaying a control worm in Fig. 3c would help prove that this is a specific fluorescent signal, and not just autofluorescence.

4. The researchers do not mention how often the fluorescence was observed, or under which conditions this was the case (they state "in certain conditions").

5. In Fig. 4 it is stated that n>6, but it is not clear to me what exactly this refers to - number of worms, number of replicates (each replicate containing how many worms?), ...

6. While Table 1 gives a comprehensive overview of all conditions tested, it isn't clear to me which combinations were tested, which might be useful information for future research. Moreover, no data or statistics are shown, it is just said that "none of the tested conditions or their combinations resulted in successfully transformed C. elegans".

7. It might be an issue with uploading the figures, but the resolution in the version I received is much too low.

I hope my feedback is useful to the authors, and they will soon be able to publish their results.

Reviewer #3: In this manuscript (PONE-D-24-38923), Warren et al. present data from their attempt to establish a platform for the genetic transformation of C. elegans using Agrobacterium. This bacterial strain is a natural plant pathogen, primarily used for genetic transformation of plants, with limited studies demonstrating its use for transformation in other kingdoms and even human cell lines. The authors argue that conventional methods for genetic transformation of C. elegans (microinjection and biolistic bombardment) lack the potential for high-throughput and timely generation of C. elegans mutant libraries. To address this limitation, the authors propose using different Agrobacterium strains, alongside specific constructs and screening conditions, as a food source for worms to develop a simpler and more efficient high-throughput transgenesis protocol. However, as the authors acknowledge, their attempts were unsuccessful. Nonetheless, they present their data to support the scientific community by providing constructs, conditions, and optimization suggestions, aiming to serve as a foundation for future successful endeavors.

Also, using established systems for phenotyping C. elegans behavior, the authors tested different constructs with genes responsible for inducing paralysis. Despite these constructs failing to induce paralysis due to unsuccessful transformation from Agrobacterium, the authors assessed worm behavior in terms of forward and paused motion frequencies.

Overall, the manuscript introduces a novel yet unsuccessful approach for gene transfer from Agrobacterium to C. elegans. Given the significance of C. elegans as a model organism in genetic studies, a contribution acknowledged by four Nobel Prizes, developing a simple and high-throughput method for genetic manipulation is of immense interest. While the authors provide an honest representation of their data, some claims require justification to enhance the reliability of the study and ensure its suitability for publication. Below, I provided some suggestions to improve the manuscript’s rigor and address gaps in study design and presentation.

Major Points:

1. The authors state in the Results section: “In certain conditions, worms fed Agrobacterium containing T-DNA designed to introduce wrmScarlet expression did display remarkable fluorescence (Fig. 3c).” Additionally, they claim: “Bright fluorescence appeared to localize to the pharynx, the vulva, and two regions on the ventral side of the nematode approximately 100 μm anterior and posterior to the vulva, consistent with the spermatheca.” Similar claims appear in the Discussion, suggesting that Agrobacterium has the potential to target the germline of C. elegans for generating heritable transformants: “Despite our lack of success, we have observed Agrobacterium triggered fluorescence in the reproductive tissues of the vulva and spermatheca, suggesting that Agrobacterium has the potential to target the germline of C. elegans for the generation of heritable transformants.” and “This indicates that Agrobacterium are interacting at these locations, to trigger fluorescence, rather than accumulating directly. This finding suggests that Agrobacterium-mediated transformation is a promising approach since Agrobacterium are able to access reproductive tissues suggesting a propensity to target the germline of C. elegans for the generation of heritable transformants.”

Although the authors mentioned that the data presented was not always reproducible, the data presented in Fig. 3c do not convincingly support these claims. The fluorescence attributed to the spermatheca is based on rough distance calculations. Additionally, the control worm in Fig. 3a shows autofluorescence comparable to the foci observed in Fig. 3c.

• To substantiate this claim, the authors should:

o Provide additional evidence, such as staining with DAPI to detect sperm cell nuclei or using differential interference contrast (DIC) optics to superimpose fluorescent channels on reproductive tissues. These methods would help verify whether the fluorescence observed corresponds specifically to the spermatheca and support the claim of targeted interaction by Agrobacterium.

o Reevaluate the conclusion that the observed fluorescence is a direct effect of GFP-expressing bacteria. Without stronger evidence, the assertion that Agrobacterium has potential for germline transformation is misleading.

2. The authors state that “C. elegans were able to survive and reproduce on A. tumefaciens, as demonstrated by the presence of progeny after 4 days.” However, no supporting data are provided. The ability to produce viable progeny is crucial for successful genetic transformation.

• It is recommended that the authors conduct progeny assays to assess the number and viability of offspring when worms are fed on A. tumefaciens compared to E. coli. Include this data as supplemental material to strengthen the study.

3. While the authors’ attempts at gene transfer were unsuccessful, including data on the general transformation efficiency of A. tumefaciens would help assess the feasibility of this approach and provide a benchmark for evaluating its potential in C. elegans. Including this information as supplemental data could help the scientific community better evaluate the feasibility of the proposed approach.

4. Table 1 lists “Conditions Screened for Agrobacterium Induction and Co-incubation with C. elegans.” While potentially useful, the table lacks important details, likely due to insufficient study design.

• A more comprehensive table, including key experimental parameters and results, would improve the manuscript’s utility for other researchers.

Minor Points:

1. Reference 14, a master’s thesis analyzing host-pathogen interactions between Agrobacterium and C. elegans, does not include experiments on genetic transformation. However, this reference (although not peer-reviewed), suggests that C. elegans can survive on A. tumefaciens as a sole food source, as shown by longevity assays.

• The current manuscript evaluates worm crawling speed and length when grown on A. tumefaciens. Since the cited reference includes viability assays, adding comparative data on worms grown on E. coli would provide better context for these observations. Additionally, examining germline morphology, cell migration, or other anatomical or physiological factors could offer deeper insights into the survival and adaptation of C. eleganswhen fed exclusively on A. tumefaciens.

Reviewer #4: This manuscript investigates the potential of using Agrobacterium tumefaciens for high-throughput genetic transformation of C. elegans. The authors demonstrate that worms can survive and reproduce using Agrobacterium as a food source, and sporadic fluorescence in reproductive tissues suggests possible germline targeting. However, routine transgenesis was not achieved. The study explores various transformation protocol parameters to facilitate gene transfer, but these efforts did not yield successful transformation.

The low throughput of transgenesis remains a significant bottleneck in C. elegans genetics. The authors' attempts, although not yet successful, are valuable to the community. Despite the lack of positive results, this manuscript provides a well-documented account of experimental procedures, optimization strategies, and a useful resource of T-binary constructs, offering insights for future attempts.

Major Comments

1. Line 99: The authors observed progeny after 4 days on A. tumefaciens. In my lab, N2 animals begin laying eggs at 3 days of age on E. coli at 20°C. Are there differences in the growth rates of animals fed on A. tumefaciens compared to E. coli? Measuring growth rates could provide a complementary metric to length and speed, offering a more comprehensive fitness assessment for animals cultured on different food sources.

2. Lines 123–126: The meaning of "in certain conditions" is unclear. Could the authors specify what these conditions entail? Describing them in detail is crucial, as the factors causing Agrobacterium to target specific regions could inform strategies for successful gene delivery.

3. Line 162: The sample size for behavioral phenotyping is too small to draw solid conclusions. I recommend increasing the sample size, ideally doubling it, and reanalyzing the data to strengthen the conclusions.

4. Lines 225-227: The authors suggest introducing male worms to enhance Agrobacterium's entry through the vulva. Running a quick experiment to test this hypothesis would provide valuable supporting evidence.

5. General Thought for Future Research: Before testing on whole animals, it might be worthwhile to perform ex vivo infection tests using cultured C. elegans germ cells. This approach could help refine conditions for optimal Agrobacterium attachment and gene delivery into cells, providing a more controlled environment for initial explorations.

Minor Comments:

1. Lines 112–113: These lines should not be divided by a new line, as they are part of the same sentence.

2. Line 143: Replace "pmyc3" with "pmyo3."

3. Line 207: A full stop is missing before "Here we"

4. Line 280: Replace "form" with "from."

6. PLOS authors have the option to publish the peer review history of their article (what does this mean? ). If published, this will include your full peer review and any attached files.

**Do you want your identity to be public for this peer review?** For information about this choice, including consent withdrawal, please see our Privacy Policy .

Reviewer #1: No

Reviewer #2: No

Reviewer #3: No

Reviewer #4: No

---

## [Author Response · Author response to Decision Letter 1]

18 Mar 2025

Dear Dr Srinivasan

Thank you for the opportunity to revise and improve our manuscript in accordance with PLOS ONE's requirements. We sincerely appreciate the thoughtful feedback provided by the reviewers and have carefully addressed each of their comments. In the rebuttal letter (Response to Reviewers.docx), we document the specific changes made in response to the four reviewers’ suggestions, for clarity our responses are in blue, and changes to the manuscript are also included in the document “Revised Manuscript with Track Changes.docx”. Some additional formatting corrections (typos, and italics etc.) have been made in the final document “Manuscript.docx”. Additionally I have uploaded and published detailed versions of the protocols on protocols.io and have included the DOI in the methods section as recommended. We hope that our responses and the revised manuscript meet the expectations of the editorial team and reviewers.

Yours Sincerely,

Eleanor Warren

---

## [Decision Letter · Decision Letter 1]

6 May 2025

Screening conditions and constructs for attempted genetic transformation of C. elegans by Agrobacterium

PONE-D-24-38923R1

Dear Dr.,

We’re pleased to inform you that your manuscript has been judged scientifically suitable for publication and will be formally accepted for publication once it meets all outstanding technical requirements.

Kind regards,

Ramachandran Srinivasan, Ph.D.

Academic Editor

PLOS ONE

---

## [Editor Report · Acceptance letter]

PONE-D-24-38923R1

PLOS ONE

Dear Dr. Warren,

I'm pleased to inform you that your manuscript has been deemed suitable for publication in PLOS ONE. Congratulations! Your manuscript is now being handed over to our production team.

Kind regards,

on behalf of

Dr. Ramachandran Srinivasan

Academic Editor

PLOS ONE